# Cardiopulmonary Effects of COVID-19 Vaccination: A Comprehensive Narrative Review

**DOI:** 10.3390/vaccines13060548

**Published:** 2025-05-22

**Authors:** Lauren T. Forchette, Luis Palma, Christian Sanchez, Rebecca M. Gibons, Christoph A. Stephenson-Moe, Benjamin J. Behers

**Affiliations:** 1Florida State University Internal Medicine Residency at Sarasota Memorial Hospital, Sarasota, FL 34239, USA; lauren-forchette@smh.com (L.T.F.); luis-palma@smh.com (L.P.); 2Department of Clinical Sciences, College of Osteopathic Medicine, Nova Southeastern University, Davie, FL 33328, USA; cs3653@mynsu.nova.edu; 3Department of Clinical Sciences, College of Medicine, Florida State University, Tallahassee, FL 32304, USA; rmg18bg@med.fsu.edu (R.M.G.); cstephensonmoe@med.fsu.edu (C.A.S.-M.)

**Keywords:** cardiovascular, pulmonary, COVID-19 vaccine

## Abstract

Coronavirus disease 2019 (COVID-19) messenger RNA (mRNA) vaccines have been associated with numerous side effects since their widespread release to the public. Cardiovascular complications include myocarditis and pericarditis, Takotsubo cardiomyopathy, postural orthostatic tachycardia syndrome (POTS), arrhythmias, sudden cardiac death, and cardiac tamponade. Pulmonary complications are pulmonary embolism (PE), interstitial lung disease (ILD), idiopathic pulmonary fibrosis (IPF), pneumonia, eosinophilic granulomatosis with polyangiitis, pneumonitis, and pulmonary hypertension. Despite these complications, the risk–benefit analysis still strongly favors vaccination, as these events occur more frequently with natural infection and confer a significantly worse prognosis. This study outlines the evidence surrounding each attributed effect, the clinical course including diagnosis and management, and the proposed pathophysiology. To our knowledge, this is the most comprehensive review of the cardiopulmonary effects of COVID-19 vaccination to date.

## 1. Introduction

Severe acute respiratory syndrome coronavirus 2 (SARS-CoV-2) is the virus responsible for the coronavirus disease 2019 (COVID-19) global pandemic, which has had hundreds of millions of infections, as well as millions of fatalities, to date [1]. At the end of 2019 and beginning of 2020, COVID-19 quickly proved to be extremely contagious with cases growing exponentially and spreading globally within a matter of months. Hospital systems around the world rapidly became overwhelmed with COVID-19 patients, with supply shortages of personal protective equipment (PPE), ventilators, medications, and virtually everything else necessary to combat a global pandemic, including healthcare staff. Due to its novelty, effective treatment strategies had yet to be developed, and various methods were subsequently trialed by physicians and researchers. Widespread lockdowns were adopted by countries, in hopes of slowing the spread of the virus. Additionally, companies across all industries began modifying their production to aid in providing pandemic supplies, such as PPE and antiseptics. Despite all these efforts, morbidity and mortality of COVID-19 remained high throughout 2020 and lockdowns remained in place for many countries.

Behind the scenes, vaccine manufacturers were working on messenger RNA (mRNA) vaccines for SARS-CoV-2. Their efforts commenced at a rapid pace, with the goal of achieving an effective vaccine as soon as possible. Technology surrounding mRNA vaccines had long been established and was adapted to target SARS-CoV-2 [1,2]. COVID-19 mRNA vaccines contain nucleoside-modified messenger RNA (mRNA), encoding the SARS-CoV-2 spike glycoprotein, encapsulated in lipid nanoparticles [2]. Cells then produce the viral spike protein, which induces an adaptive immune response [2]. Resultant spike-protein IgG antibodies prevent viral attachment to angiotensin-converting enzyme 2 (ACE2) receptors on the host cell, thereby inhibiting entry and neutralizing the virus [2]. Alongside the production of these antibodies is a significant inflammatory response via cytokine activation [1]. This inflammatory response leads to the local and systemic side effects attributed to these vaccines.

Development of these mRNA vaccines proved to be a turning point in the pandemic. Following positive study results, the first COVID-19 mRNA vaccine, Pfizer-BioNTech’s BNT162b2 (Pfizer), received emergency use authorization (EUA) from the United States’ Food and Drug Administration (FDA) on 11 December 2020 [1]. The Pfizer vaccine trial consisted of 43,548 total participants with 21,720 receiving the vaccine and 21,728 receiving placebo. The vaccine conferred 95% protection against infection from COVID-19 and similar efficacy across age, sex, race, ethnicity, baseline body-mass index (BMI), and coexisting condition subgroups [3]. Furthermore, the incidence of serious adverse events was low, while self-limiting side effects of mild-to-moderate pain at the injection site, fatigue, and headache were similar to other viral vaccines, albeit at a 2-month median follow-up [3]. Similarly, Moderna’s mRNA-1273 (Moderna) received EUA from the FDA on 18 December 2020 [1]. Its trial enrolled 30,420 participants with 15,210 in both the vaccine and placebo groups, with the vaccine conferring 94.1% efficacy at preventing COVID-19 infection and similar efficacy also observed across subgroups [4]. Additionally, no safety concerns were identified with the Moderna vaccine, aside from transient local and systemic reactions [4]. Following EUA, countries sought widespread dissemination of these vaccines to the general public, with billions of doses being received globally within the first six months [1]. As a result of widespread immunity, lockdowns began to abate, and the world began to turn the page on the COVID-19 pandemic.

However, although only mild side effects were noted in the mRNA vaccine trials, reports of more serious adverse events began to surface following their widespread use amongst the public. This study will focus on some of the cardiopulmonary effects attributed to COVID-19 vaccination. We will outline the evidence surrounding each attributed effect, the clinical course including diagnosis and management, and the proposed pathophysiology.

## 2. Methods

We searched PubMed for articles surrounding cardiopulmonary effects of COVID-19 vaccination as of 7 May 2025. Our search terms were “cardiac effects COVID vaccine” and “pulmonary effects COVID vaccine”, yielding 1603 and 2514 results, respectively. Studies of interest were those documenting the incidence and/or clinical course of cardiopulmonary complications from mRNA COVID-19 vaccines. Abstracts of all articles identified in the search were reviewed for their adherence and potential contribution to our study. In instances where multiple similar studies existed, those with the larger sample size were utilized in this review. Reference lists of included studies were also reviewed to identify additional studies for inclusion.

## 3. Cardiovascular Effects of COVID-19 Vaccination

### 3.1. Myocarditis and Pericarditis

Myocarditis and pericarditis are the most notorious and extensively studied cardiovascular effects associated with COVID-19 vaccines, particularly mRNA vaccines such as Pfizer and Moderna. The risk of myocarditis from these mRNA vaccines has been estimated at 4–28 cases per 100,000 doses, with increased risk seen in males, people under 30 years old, and after the second dose [5]. Notably, the risk of myocarditis following COVID-19 vaccination is less than that conferred through natural infection, with a recent meta-analysis estimating the incidences at 19.7 per 1,000,000 and 2.76 per 1000, respectively [6].

Patients with COVID-19 vaccination-associated myo-/pericarditis typically present three days following vaccination with chest pain, fever, and dyspnea [5,7]. Common laboratory findings include elevated troponin, B-type natriuretic peptide (BNP), and inflammatory markers, such as c-reactive protein (CRP) and erythrocyte sedimentation rate (ESR) [7]. Electrocardiogram (EKG) abnormalities are seen in 80–90% of cases with ST elevations, sinus tachycardia, and nonspecific ST-segment and T-wave changes being the most common [5,7,8]. Echocardiogram (ECHO) findings include left ventricular ejection fraction (LVEF) below 50% in anywhere from 21 to 44% of cases, while pericardial effusions are seen approximately 20% of the time [5,7,8]. Definitive diagnosis of myo-/pericarditis is achieved either through cardiac magnetic resonance imaging (cMRI), which would show late gadolinium enhancement corresponding to myo- or pericardial inflammation, or through cardiac biopsy, which is utilized less frequently due to its invasive nature [7,8].

Treatment of this phenomenon is typically supportive with anti-inflammatories, such as nonsteroidal anti-inflammatory drugs (NSAIDs), aspirin, colchicine, and steroids [7,8]. Guideline-directed medical therapy with beta-blockers, angiotensin-converting enzyme (ACE) inhibitors or angiotensin II receptor blockers (ARBs), and diuretics are also frequently used, particularly in cases with reduced LVEF [8]. Fortunately, myo-/pericarditis associated with COVID-19 vaccination has a favorable prognosis with resolution of symptoms in most cases and median hospitalizations around 3–5 days [7,8]. However, one study of 182 patients noted that three cases (1.6%) resulted in death, so severe adverse outcomes are possible [8]. Additionally, persistently reduced LVEF can be seen in around 10% of cases, while ongoing symptoms have been seen in as many as 12% of cases [5]. Furthermore, persistent abnormalities on cMRI are seen in around 50% of cases at a median follow-up time of 6–7 months [5]. In comparison to myo-/pericarditis from natural COVID-19 infection, vaccine-associated cases appear to have a more benign disease course with less need for invasive treatment, increased rates of LVEF recovery to baseline, and significantly fewer mortalities [5,7].

Despite its well-documented nature, the underlying mechanism behind this association is still a topic of debate. Hypotheses include activation of the immune system, hypersensitivity reactions, and molecular mimicry [9]. Activation of the innate and adaptive immune systems occurs in response to either the spike glycoprotein or the mRNA itself, resulting in cardiac inflammation through upregulation of pro-inflammatory pathways and cytokines [5,9]. Hypersensitivity reactions are also possible because of an immune response to the other components of the vaccine, such as the lipid nanoparticle, whose ionizable lipids can activate toll-like receptors and lead to myocarditis [5]. Furthermore, molecular mimicry can occur when antibodies against the spike protein cross-react with cardiac self-antigens, such as myosin heavy chains or troponin C1, causing inflammation [4,5]. Additionally, higher levels of androgens may enhance the pro-inflammatory responses of the immune system, while estrogen exhibits an anti-inflammatory effect, highlighting a possible explanation for why this phenomenon is seen more frequently in males [4,5].

### 3.2. Takotsubo Cardiomyopathy

Takotsubo cardiomyopathy (TCM), also known as broken heart syndrome and stress-induced cardiomyopathy, has increased in prevalence since the start of the COVID-19 pandemic, being attributed to both natural infection and vaccines [10]. TCM is characterized by a transient left ventricular dysfunction which is often precipitated by emotional or physical stress. TCM following COVID-19 vaccination tends to occur in older females, with systematic reviews yielding a median age of 61.5 years and female predominance as high as 75–90% [11,12]. Patients typically present 2–3 days following vaccination with chest pain and dyspnea, while hospitalization lasts an average of 7–10 days [11,12]. Elevated troponin and abnormal EKG findings are seen in all patients, while ECHO findings are variable with reduced LVEF seen in anywhere from 10 to 90% of patients [11,12]. Although there are no guidelines or protocol, treatment typically includes heart failure medications, such as ACE inhibitors or ARBs and beta-blockers [10]. Mortality and recurrence rates are estimated at 3.5–10.6% and 2–11%, respectively [10].

Despite the pathophysiology being not entirely understood, multiple hypotheses surrounding TCM from COVID-19 vaccination exist. One hypothesis suggests myocardial stunning from microvessel or multi-vessel vasospasm and direct myocardial injury in response to elevated catecholamine levels seen during stress [11]. Another involves elevated pro-inflammatory cytokines, as seen following vaccination, causing this phenomenon [11]. Furthermore, the interaction between spike proteins and ACE2 receptors may cause a relative overactivity of the pro-inflammatory angiotensin II [11].

### 3.3. Postural Orthostatic Tachycardia Syndrome (POTS)

Postural orthostatic tachycardia syndrome (POTS) is a form of dysautonomia that has been linked to COVID-19 vaccinations, although the odds of development are 1.52–2.12 times more likely from natural infection [13,14]. POTS is characterized by an excessive heart rate increase after standing, with associated dizziness, fatigue, and palpitations [15]. Demographics of patients developing POTS following COVID-19 vaccination are 59% female with a mean age of 56, with 67% Caucasian, 12% Hispanic, 11% African American, and 9% Asian [13]. Patients typically present within a few days of vaccination and the diagnosis is clinical, although elevated norepinephrine levels, reduced heart rate variability, and a positive tilt table test can be seen [15]. Treatment typically involves lifestyle modifications, such as increased fluid and salt intake, exercise, and compression stockings, although pharmacological therapy with ivabradine, corticosteroids, and beta-blockers, amongst other medications, can also be tried [15]. Studies have shown that symptoms could persist for months in some cases, especially in patients with a history of autonomic dysfunction [16]. Possible mechanisms surrounding this association include immune-mediated transient autonomic dysfunction, molecular mimicry between vaccine antigens and autonomic pathways, or dysautonomia from vaccine-induced inflammation, similar to the mechanism underlying post-viral autonomic syndromes [15].

### 3.4. Arrhythmias

Arrhythmias, ranging from benign palpitations to life-threatening conditions, have been reported following COVID-19 vaccination. A large meta-analysis determined an incidence of 22 per 10,000 people following the Pfizer vaccine and 76 per 10,000 following Moderna [17]. Both bradyarrhythmias and tachyarrhythmias have been observed. Complete AV block has been reported after patients presented with syncope and dizziness within a few weeks of COVID-19 vaccination, with treatment via pacemaker insertion [18]. Tachyarrhythmias include atrial fibrillation (AF) and other supraventricular tachycardias, ventricular tachycardia (VT), and ventricular fibrillation (VF). Atrial fibrillation has been observed at an incidence of 5 cases per million doses of COVID-19 vaccine, particularly in older adults or those with prior cardiovascular comorbidities, with patients presenting within a week or two of vaccination [19]. COVID-19 vaccination has also been implicated in unmasking Brugada in genetically predisposed individuals, due to fever-induced sodium channel dysfunction, presenting as VT [20,21,22]. Similarly, vaccination can unmask underlying Long QT syndrome and present as syncope with VT [23,24]. Additionally, isolated cases of VT and VF following vaccination have been documented, often in individuals with underlying structural heart disease or concomitant myocarditis [25,26]. Treatment and prognosis are arrhythmia-specific, as are proposed mechanisms. In short, bradyarrhythmias are attributed to the humoral response from COVID-19 vaccinations, inducing conduction system abnormalities, atrial tachycardias are probably from underlying pro-inflammatory cytokines and molecular mimicry, while ventricular tachycardias are associated with myocardial edema and ischemia [18].

### 3.5. Sudden Cardiac Death

Sudden cardiac death (SCD) is an exceptionally rare but devastating adverse event reported following COVID-19 vaccination [27]. Most documented cases have been associated with myocarditis or undiagnosed pre-existing cardiac conditions, both of which can lead to fatal arrhythmias [27,28]. Mechanistically, SCD typically results from a similar pathophysiology to the previously mentioned cardiovascular complications since it is usually secondary to another underlying cardiac process. Autopsy findings in reported cases frequently reveal significant inflammatory infiltration of myocardial tissue, consistent with vaccine-associated myocarditis [29]. The temporal association between COVID-19 vaccination and SCD ranges from days to a few weeks after vaccine administration, stressing the need for an increased clinical vigilance in at-risk populations [30]. For example, individuals with predisposing factors such as prior myocarditis, genetic arrhythmia syndromes, or significant structural heart disease may require careful monitoring. Despite being such a rare adverse event, SCD has profound implications with a high risk of mortality, emphasizing the importance of balancing the benefits of vaccination against these risks and tailoring recommendations for high-risk groups. Detailed case reports have highlighted the importance of early recognition of warning signs such as chest pain and syncope, which could precede these fatal arrhythmias [31].

### 3.6. Cardiac Tamponade

Cardiac tamponade has also been reported in a few isolated cases following COVID-19 vaccination [32,33]. In these reports, patients often present with dyspnea, hypotension, and jugular venous distension within days to weeks of vaccination [32,33]. The condition can arise secondary to severe pericarditis or as part of a broader inflammatory response [34]. Echocardiographic findings typically show a pericardial effusion with some evidence of hemodynamic compromise such as right atrial or right ventricular diastolic collapse, as well as microthrombi formation [32,35]. Pericardiocentesis fluid tends to yield inflammatory exudates suggestive of immune-mediated pericardial injury [32]. Emerging reports suggest that persistent effusions may occur in rare cases, requiring repeated interventions and long-term anti-inflammatory therapy [35]. Recognizing at-risk individuals, particularly those with pre-existing pericardial conditions, is crucial for well-timed management.

### 3.7. Other Reported Cardiovascular Events

In addition to the conditions described above, there are other rare cardiac events that have been associated with COVID-19 vaccination. Acute coronary syndrome (ACS) has been reported in individuals with predisposing risk factors, likely triggered by heightened inflammatory states post-vaccination [36]. Hypertensive crises and exacerbation of pre-existing cardiac conditions, such as heart failure and atrial fibrillation, have also been noted in temporal association with the vaccination [37]. Differentiating these events from just merely coincidental occasions is important, given the large number of people receiving the vaccine worldwide.

A summary of the cardiovascular effects of COVID-19 vaccination and their management can be found in Table 1.

### 3.8. Comparative Risk Analysis

When considering the pros and cons in terms of the cardiovascular effects of COVID-19 vaccination, it is important to compare these risks with those imposed by natural SARS-CoV-2 infection. As alluded to above, natural COVID-19 infection has been associated with a substantially higher risk of myocarditis, arrhythmias, thromboembolic events, and other cardiac complications [38,39]. Large meta-analyses exist comparing the rates of myocarditis from vaccination to those following natural infection. One such study consisting of 55.5 million vaccinated individuals and 2.5 million with natural infection found the relative risk of myocarditis to be more than seven times higher in the natural infection group [38]. Another study determined the incidence to be 2.76 per thousand in the natural infection group compared to 19.7 per million in the vaccine group [6]. Furthermore, they noted the vaccine group had lower mortality and rates of cardiogenic shock, as well as less frequent use of more advanced therapies, including mechanical circulatory support [6]. However, the mean age was younger in the vaccine group, while the proportion of males was also larger in that group, highlighting the increased incidence of vaccine-associated myocarditis in young males [6].

The data directly comparing other cardiovascular complications between COVID-19 vaccination and natural infection are less prevalent and thus must be extrapolated. For instance, the rates of TCM cases increased from 1.5% to 7.8% during the COVID-19 pandemic with only ten published case reports attributed to vaccination at the time, implicating the role of natural infection in a substantial number of cases [11]. Another study of TCM cases during the COVID-19 pandemic found 66.7% of patients had concomitant COVID-19 infection [10]. Comparison of vaccine-associated TCM to patients hospitalized with TCM from any cause found all vaccine cases recovered without complications over a mean hospitalization of 10 days, compared to an in-hospital mortality rate of 1.3%, discharge home rate of 73.6%, and 1-year mortality of 6.9% [11]. These findings highlight that vaccine-associated TCM is both exceedingly rare and confers a better prognosis than TCM overall.

Natural COVID-19 infection has also been shown to be strongly associated with cardiac arrhythmias and autonomic dysfunction, such as POTS. A study of 81,844 patients with COVID-19 infection found sinus bradycardia in 1.9% and complete heart block in 0.01% of patients [40]. Additionally, studies have shown rates of atrial fibrillation in hospitalized patients with natural infection of 13% to 17% and that the presence of atrial fibrillation is associated with higher mortality [40]. Furthermore, rates of ventricular arrhythmias in patients with natural infection have been estimated at between 0.1% and 8% and are also associated with increased mortality [40]. These numbers are far greater than the incidence of vaccine-associated arrhythmias mentioned above of 22 per 10,000 people following the Pfizer vaccine and 76 per 10,000 following Moderna [17]. Meanwhile, postacute sequalae of SARS-CoV-2 has been reported in approximately 10% of patients following COVID-19 infection and POTS has been shown to occur in 25% to 35% of these patients [40]. As also mentioned above, the odds of the development of POTS are 1.52–2.12 times more likely from natural infection than vaccination [13,14].

This comparative analysis shows the incidence of all studied cardiovascular complications to be higher in the setting of COVID-19 natural infection than from vaccination. Furthermore, the data surrounding clinical course and outcomes show a better prognosis when these cardiovascular complications are the result of vaccination rather than natural infection. These findings support the relative safety of vaccination despite the associated complications. Public health efforts should emphasize the rarity of vaccine-associated events in contrast to the documented burden of cardiovascular sequelae from COVID-19.

## 4. Pulmonary Effects of COVID-19 Vaccination

### 4.1. Pulmonary Embolism

Although vaccine-induced thrombosis and thrombocytopenia (VITT) is a rare complication primarily linked to adenoviral vector vaccines, cases of pulmonary embolism (PE) have also been documented following mRNA COVID-19 vaccination [41]. A systematic review identified 301 cases of PE among 17,636 cardiovascular events reported after mRNA vaccination [42]. Similarly, another study found that while the risk of adverse events post-mRNA vaccination remains low, PE and deep vein thrombosis (DVT) were among the most frequently reported thrombotic complications [43]. Numerous case reports exist documenting this phenomenon, typically days to weeks following vaccination, and often in individuals with no prior thrombotic risk factors [44,45,46,47,48].

COVID vaccine-associated PE typically presents with dyspnea, chest pain, and palpitations, while cough and syncope have also been reported [44,47]. Cardiac arrest requiring resuscitation is a feared complication reported in some case reports [45,47]. Diagnosis is established primarily through computed tomography (CT) pulmonary angiography, which shows clot burden in the pulmonary arteries [45,46]. Cardiac imaging findings include right ventricular dilation and pulmonary hypertension [46]. Elevated c-reactive protein (CRP) levels can also be seen, highlighting underlying inflammation [45,46,48]. Treatment consists predominantly of thrombolytic therapy, low-molecular-weight heparin (LWMH), and Factor Xa inhibitors, such as apixaban and rivaroxaban, leading to favorable outcomes in most cases [44,45,46,47]. However, persistent pulmonary hypertension and right heart strain can be seen in some patients [46]. Furthermore, patients with pre-existing pulmonary disease may experience severe complications, including respiratory failure and mortality [48].

The pathophysiology of vaccine-associated PE remains unclear. The proposed mechanisms include platelet factor 4 (PF4) activation, leading to a hypercoagulable state similar to VITT [49]. Another hypothesis is cytokine-induced systemic inflammation causing concomitant endothelial damage and a prothrombotic state, which predisposes to thrombus formation with embolic travel to the pulmonary vasculature [48]. Future studies into this phenomenon are needed to further elucidate the true mechanism and determine whether causation exists between COVID-19 vaccines and thrombotic events.

### 4.2. Interstitial Lung Disease

Interstitial lung disease (ILD) has emerged as a rare but notable pulmonary complication following COVID-19 vaccination, with several studies assessing the incidence, potential mechanisms, and clinical outcomes associated with vaccine-related ILD exacerbations. While large-scale analyses have not demonstrated a statistically significant increased risk of ILD following vaccination, isolated case reports and retrospective studies suggest that certain subgroups, particularly those with pre-existing autoimmune-related ILD could be at a higher risk for exacerbations. A retrospective study of 545 ILD patients found that 17 individuals (3.1%) experienced worsening respiratory symptoms following COVID-19 vaccination, with 4 cases (0.7%) meeting criteria for acute exacerbations of ILD (AE-ILD) requiring hospitalization [50]. Further large-scale disproportionality analysis using VigiBase, a global pharmacovigilance database, identified 679 cases of ILD among 1,233,969 vaccine-related reports, predominantly linked to mRNA vaccines such as Pfizer (55.4%) and Moderna (11.5%) [51]. These cases were managed with steroid pulse therapy, while two patients required additional immunosuppressive treatment with IV cyclophosphamide [50]. Although AE-ILD post-vaccination is rare, vigilance is necessary, particularly in patients with autoimmune-related ILD. Drug-induced ILD (DI-ILD) has also been reported following COVID-19 vaccination, with cases presenting with acute respiratory failure, hypoxemia, and extensive ground-glass opacities on imaging [52]. Two cases of DI-ILD involved elderly male patients developing acute respiratory distress within days of vaccination, with laboratory findings of elevated pulmonary injury markers and inflammatory cytokines [52]. Both cases demonstrated clinical improvement following corticosteroid therapy, reinforcing the importance of early recognition and intervention [52].

The immunopathogenesis underlying vaccine-associated ILD remains speculative, with many different proposed mechanisms. One mechanism involves exaggerated immune activation through cytokine release, particularly interleukin (IL)-2, tumor necrosis factor-alpha (TNF-α), and interferon-gamma (IFN-γ), which could lead to alveolar inflammation [53]. Additionally, mRNA vaccine-induced Th1-cell responses may promote macrophage activation, exacerbating pre-existing pulmonary fibrosis [53]. Vaccine adjuvants have also been implicated in immune dysregulation, potentially triggering inflammatory cascades leading to pneumonitis or fibrotic lung injury [53]. Further research into this pathophysiology is warranted to further the risk–benefit analysis of vaccination in this population.

Despite these concerns, emerging evidence suggests that COVID-19 vaccination may not only be safe for ILD patients but may also confer a protective effect against ILD development. A retrospective cohort study utilizing a Korean national database analyzed over 1.1 million matched individuals and found a significantly lower incidence of ILD among vaccinated individuals compared to their unvaccinated counterparts [54]. The incidence rate was 6.8 per 10,000 person-years in the vaccinated group versus 10.6 per 10,000 in the unvaccinated group (*p* < 0.0001), suggesting that COVID-19 vaccination may reduce the overall burden of ILD at the population level [54].

### 4.3. Idiopathic Pulmonary Fibrosis

As with ILD, COVID-19 vaccination has been associated with acute exacerbations of idiopathic pulmonary fibrosis (IPF). One study on IPF patients admitted for respiratory deterioration found 10 diagnosed with acute exacerbations of IPF (AE-IPF), and 4 of those patients (40%) had received the Pfizer vaccine within the preceding three to five days [55]. Notably, two of these patients ultimately succumbed to their exacerbations [55]. However, the rest of the associations come primarily through case reports. One case involved an 82-year-old man who developed progressive dyspnea, cough, and anorexia 1.5 months post-vaccination [56]. Another case documented an 84-year-old male with fibrotic ILD who developed AE-IPF within nine days of receiving the second dose of the Pfizer vaccine [57]. A third case involved a 72-year-old male with a well-documented history of IPF who presented with respiratory decline one-week post-vaccination [58]. Diagnosis is obtained using high-resolution computed tomography (HRCT), which reveals new ground-glass opacities and honeycombing, and through bronchoscopy [56,57]. High-dose corticosteroid therapy resulted in improvement in symptoms for all patients in these case reports [56,57,58]. Despite these concerns, COVID-19 vaccination in patients with chronic lung disease to protect from severe infection is felt to outweigh the risks, but there is need for careful post-vaccination monitoring for acute exacerbations in IPF patients [59].

While causality remains unproven, several mechanisms have been proposed to explain the potential link between COVID-19 vaccination and AE-IPF. Vaccine-induced immune activation may worsen pulmonary fibrosis through excessive Th1-mediated inflammation and cytokine release [55]. The upregulation of IL-6 and IL-22 observed in IPF patients post-vaccination suggests a persistent pro-fibrotic environment, which could heighten susceptibility to disease exacerbation [55]. Additionally, alterations in immune cell populations, particularly the increase in regulatory T cells and cytotoxic lymphocytes, may play a role in modulating immune responses in a way that predisposes certain individuals to AE-IPF following vaccination [55].

### 4.4. Pneumonia

Pneumonia, both as a direct inflammatory reaction and as a complication of preexisting interstitial lung disease, has been reported in association with COVID-19 vaccination. A case of concurrent acute exacerbation of idiopathic nonspecific interstitial pneumonia (iNSIP) and pulmonary embolism was reported in an 82-year-old woman who had been clinically stable for three years before developing acute dyspnea two days after receiving a booster dose of the Pfizer-BioNTech (BNT162b2) vaccine [48]. Despite initial management, her respiratory failure worsened, eventually leading to hemodynamic instability, multiorgan failure, and death [48]. Cases of organizing pneumonia following COVID-19 vaccination have also been reported, with patients presenting with cough, dyspnea, and fever [60,61]. Two additional cases of acute exacerbation of interstitial pneumonia following mRNA COVID-19 vaccination have been reported [62]. In the first case, an 83-year-old man with stable idiopathic interstitial pneumonia developed high fever and dyspnea one day after receiving his first Pfizer vaccine dose [62]. CT imaging revealed newly developed diffuse ground-glass opacities, and laboratory tests showed elevated CRP and surfactant protein-D (SP-D), both markers of pulmonary inflammation [62]. He was treated with corticosteroid pulse therapy, leading to gradual improvement [62]. In the second case, a 65-year-old man with a history of acute exacerbation of interstitial pneumonia, previously treated with steroids, developed low-grade fever and dyspnea six days after receiving his second Pfizer vaccine dose [62]. CT imaging revealed ground-glass opacities with traction bronchiectasis, while blood tests showed severe anemia, leukopenia, and elevated CRP and SP-D [62].

Mechanisms underlying vaccine-related pneumonia is not completely understood. Vaccine-induced immune activation may play a role through excessive Th1-mediated inflammation and cytokine release, leading to increased IL-6 and tumor TNF-α production, which are known mediators of lung injury [62]. Additionally, molecular mimicry between vaccine antigens and pulmonary self-antigens could theoretically contribute to autoimmune-like reactions, particularly in patients with preexisting ILD or immune dysregulation [62]. Similar underlying cytokine storm reactions have been linked to severe complications from natural COVID-19 infections, which typically have a worse prognosis than its vaccine counterparts [48].

### 4.5. Eosinophilic Granulomatosis with Polyangiitis

Eosinophilic granulomatosis with polyangiitis (EGPA), formerly known as Churg–Strauss syndrome, is a rare systemic vasculitis characterized by eosinophilic inflammation, asthma, and multiorgan involvement [63]. It is part of the spectrum of antineutrophil cytoplasmic antibodies (ANCA)-associated vasculitides and is thought to result from immune dysregulation leading to excessive eosinophilic activation and vascular inflammation [63]. Several cases of new-onset EGPA and related vasculitides, including granulomatosis with polyangiitis (GPA), have been reported following COVID-19 mRNA vaccination, raising concerns about potential vaccine-induced immune activation in predisposed individuals.

Case reports documenting this highlight patients presenting with a variety of nonspecific symptoms, such as low-grade fever, edema, myalgias, nasal congestion, and rash 2–15 days following COVID-19 vaccination [64,65,66,67,68]. Laboratory findings include severe eosinophilia, elevated inflammatory markers, and elevated myeloperoxidase (MPO) antibodies [64,65,66,68]. Treatment is primarily corticosteroids and cyclophosphamide for more severe cases, while plasmapheresis, rituximab, and tumor necrosis factor (TNF) inhibitors can be used for refractory disease [63].

Cases of new-onset EGPA and GPA following COVID-19 vaccination raise important questions about the underlying immunological mechanisms. Several potential explanations have been proposed, including molecular mimicry, bystander activation, and immune system priming [64,65,66,67]. Molecular mimicry occurs when vaccine components resemble self-antigens, triggering an autoimmune response [64,65]. Bystander activation involves excessive immune stimulation leading to collateral tissue damage [64]. The presence of ANCA in multiple post-vaccination cases suggests that the vaccine may trigger an underlying immune response leading to EGPA in predisposed individuals [66].

### 4.6. Pneumonitis

Pneumonitis is a non-infectious inflammatory reaction of the lung parenchyma and has been reported in rare cases following COVID-19 vaccination. A large-scale nationwide multicenter survey conducted in South Korea identified 49 cases of COVID-19 vaccine-associated pneumonitis, occurring within a few weeks of vaccination [69]. The median age of affected individuals was 66 years, with a predominance of male patients (61%) [69]. Most cases were associated with mRNA vaccines, with the Pfizer vaccine accounting for 57% and the Moderna vaccine for 35% [69]. The most common presenting symptoms were dyspnea (87%), cough (67%), and fever (37%) [69]. HRCT scans revealed bilateral ground-glass opacities in 85% of cases, as well as interstitial infiltrates and consolidations [69]. Pulmonary function tests demonstrated a restrictive ventilatory pattern with reduced diffusion capacity for carbon monoxide, indicating impaired gas exchange [69]. Bronchoalveolar lavage analysis revealed lymphocytic alveolitis, supporting an immune-mediated pathogenesis [69]. The treatment was primarily systemic corticosteroids, and, despite therapy, four patients (9%) required mechanical ventilation and eight (17%) succumbed to their illness [69]. These findings show the potential severity of vaccine-associated pneumonitis.

Another distinct form of vaccine-related pneumonitis is radiation recall pneumonitis (RRP), a rare inflammatory reaction occurring within previously irradiated lung tissue following exposure to triggering agents such as chemotherapy, immunotherapy, or vaccines [70]. A case of RRP was reported in a 48-year-old male with locally advanced unresectable non-small-cell lung cancer, who had undergone prior chemoradiotherapy and maintenance therapy with durvalumab [70]. He received his first dose of the Pfizer vaccine eight days after his last durvalumab infusion and the second dose 21 days later [70]. He developed a fever and dry cough 19 days after the second dose of vaccine and a CT scan showed new infiltrates consistent with RRP [70].

The underlying mechanisms linking COVID-19 vaccination to pneumonitis are not yet fully understood, but several hypotheses have been proposed. Vaccine-induced immune activation may provoke excessive inflammatory responses in susceptible individuals, particularly those with preexisting ILD or prior lung-directed therapies [69]. Molecular mimicry between vaccine components and lung-specific antigens could potentially trigger an autoimmune reaction, leading to pulmonary inflammation [69]. Additionally, mRNA vaccines have been shown to induce robust cytokine responses, which may contribute to immune-mediated lung injury in predisposed individuals [69]. Similarly, RRP is felt to arise from the vaccine-induced inflammatory response within previously irradiated lung tissue [70].

### 4.7. Pulmonary Hypertension

Pulmonary hypertension following COVID-19 vaccination has been reported in a few case reports. Patients typically present with dyspnea and fatigue within days to weeks following vaccination [71,72,73]. Some cases are associated with concomitant pulmonary embolism [71,73]. Echocardiography can suggest pulmonary hypertension and note right heart strain, but definitive diagnosis requires right heart catheterization [71,72,73]. Treatment depends on the underlying cause with anticoagulation for thrombus-associated PH and nitric oxide for idiopathic [71,72,73]. Prognosis is typically poor with permanent organ damage and/or death being common [72,73].

Potential mechanisms linking COVID-19 vaccination to PH remain speculative but include endothelial dysfunction, microvascular thrombosis, and immune-mediated vascular remodeling [71,72,73]. The spike protein encoded by mRNA vaccines has been shown to interact with endothelial cells, leading to increased expression of inflammatory cytokines, endothelial activation, and a prothrombotic state [71,72]. In predisposed individuals, these changes may contribute to vascular remodeling and increased pulmonary vascular resistance, ultimately resulting in PH [73]. In cases involving microvascular thrombosis, as suggested by elevated D-dimer levels, the immune response to vaccination may trigger localized clot formation, leading to impaired pulmonary circulation and increased pulmonary pressures [71,73].

A summary of the pulmonary effects of COVID-19 vaccination and their management can be found in Table 2.

### 4.8. Comparative Risk Analysis

As with the cardiovascular complications of COVID-19 vaccination, comparison of the risks of these pulmonary effects from natural infection is important. Unfortunately, no studies were identified directly comparing the incidences or clinical courses between vaccination and natural infection. However, a large study of 792,010 vaccinated patients found 793 cases of venous thromboembolism (VTE) following vaccination for an incidence of 0.10% [74]. On the other hand, a large meta-analysis of 48 studies determined the rate of VTE amongst patients hospitalized with natural infection to be 17.3% with rates of DVT and PE of 12.1% and 7.1%, respectively [75]. Furthermore, a study of over 20 million people showed that vaccination was associated with reduced risks of VTE in both the acute (0–30 days) and post-acute phases (31–365 days) following natural infection [76]. These findings highlight the significantly increased risk of VTE from natural infection as compared to vaccination, with the latter also showing an ability to decrease the risk of occurrence in the setting of natural infection.

Despite the associations between COVID-19 vaccination and acute exacerbations of ILD and IPF, the benefit that vaccination provides for those with already compromised lungs is invaluable. Studies show that patients with ILD have an increased risk of both infection with COVID-19 and worse outcomes, with higher rates of requiring oxygen therapy, intensive care unit (ICU) admission, and mechanical ventilation, as well as a higher morality than patients without ILD [77]. Similar results have been shown in IPF patients with COVID-19, as a meta-analysis revealed a mortality rate of 34% and ICU admission rate of 31% [78]. Vaccination has been shown to reduce the risk of severe COVID-19 infection in patients with pre-existing lung disease, including ILD and IPF [79]. Furthermore, severe COVID-19 infection has been associated with the development of IPF [80]. These findings highlight the importance of vaccination in patients with ILD and IPF.

Although an extremely rare complication of COVID-19 vaccination, pneumonia and pneumonitis are hallmarks of severe natural infection, occurring in approximately 15% of patients [81]. Studies on patients hospitalized with COVID-19 estimate that 15–30% will develop severe COVID-19 pneumonia and acute respiratory distress syndrome, which are associated with significant morbidity and mortality [82]. Similarly, COVID-19 pneumonitis carries a significant risk of mortality, as well as long-term risks of lung damage and symptoms in up to 45% and 85% of patients, respectively [83]. Given that vaccination has been shown to reduce severity of infection and subsequently the risk of pneumonia and pneumonitis, the risk–benefit analysis continues to favor vaccination.

While only seen in case reports following vaccination, studies have estimated the incidence of pulmonary hypertension from natural infection as high as 15% and it is associated with more severe infection [84]. On the other hand, EGPA was only reported in case reports for both COVID-19 vaccination and natural infection, alluding to its rarity in both instances [85].

Overall, this comparative analysis suggests that the pulmonary complications of COVID-19 vaccination occur more frequently with natural infection. Furthermore, many of these complications are the result of inflammatory responses following either vaccination or infection, with severe infection being the greater risk factor of the two. Given the efficacy of vaccination in reducing severe infection, the risk–benefit analysis clearly favors vaccination. However, unlike the cardiovascular complications, the clinical courses between cases of these pulmonary complications are essentially the same between both groups. This highlights the potential severity that these complications can pose, regardless of the inciting factor.

## 5. Discussion

Our study provides a comprehensive narrative review of the cardiopulmonary effects associated with COVID-19 vaccination, including the evidence surrounding each association, clinical course, and proposed pathophysiology. With the ongoing use of COVID-19 vaccines, this study provides an important reference for public health policy and vaccination recommendations. Even though the examined cardiopulmonary effects are seen both with vaccination and natural infection, our findings favor vaccination, as the incidence and clinical severity tend to be higher with natural infection. Additionally, vaccination can be protective against some of the pulmonary complications, including VTE and the development of IPF from natural infection.

Although vaccination is favored generally, it is important to identify populations in which certain complications occur more frequently, such as myo-/pericarditis in young males, TCM in elderly females, arrhythmias and SCD in those with underlying cardiac disease, and exacerbations of chronic lung disease in those with ILD or IPF. Despite these known associations, the identification of individuals at risk of developing these complications remains an on-going area of research. In fact, the International Network of Special Immunization Services was formed specifically to further study rare complications of COVID-19 vaccines, such as myocarditis and VITT [86]. With a better understanding of which populations are at higher risk of vaccine complications, more tailored vaccination recommendations can be targeted towards these individuals.

While specifics regarding at-risk populations remains unknown, variations in cardiovascular complications have been observed with vaccine type and dose number. A large Bayesian multivariate meta-analysis found that mRNA COVID-19 vaccines were associated with increased odds of coronary artery disease (CAD) following the second dose [87]. Subgroup analyses yielded a significant increase in arrhythmia following the first dose of vaccine and an increase in myocardial infarction (MI) and cardiovascular disease risk following the second dose [87]. Additionally, the Pfizer vaccine was associated with an increased risk of CAD and MI [87]. Furthermore, aside from the negative consequences, Moderna was more effective than Pfizer in preventing COVID-19 hospitalizations in patients with underlying medical conditions [88]. Future studies into the variations in cardiovascular complications and efficacy across vaccine type and dose number are needed to confirm these findings. These could also potentially specify which vaccines are most appropriate for various populations, specifically those at risk of cardiovascular disease.

Although studies surrounding biomarkers for individuals at risk of vaccine complications are not currently available, inferences can be drawn from studies on the immune response to vaccination. One such study showed higher levels of the pro-inflammatory cytokines IL-6, CRP, and TNF-α in vaccinated individuals compared to their nonvaccinated counterparts [89]. Elevations in these cytokines were associated with a lower risk of adverse outcomes during infection, representing a controlled immune response [89]. However, at a certain level, these inflammatory markers become damaging and can represent the cytokine storm presumed to cause many of the cardiopulmonary effects of vaccination that we explored [89]. Identifying patients who will experience these significant increases in pro-inflammatory cytokines following vaccination represents an important endeavor in combating the negative effects associated with vaccination. Despite an assumed similar pathophysiology for cardiopulmonary effects of natural infection and vaccination, a recent study on myocarditis across the two entities suggested otherwise. This study obtained left ventricular endomyocardial biopsies from both natural COVID-19 infection and vaccination myocarditis, with IFN-γ being predominant in natural infection, while IL-16 and IL-18 were in vaccination [90]. Additionally, the vaccine group had a higher proportion of CD4^+^ T cells, while the natural infection group had higher levels of CD8^+^ T cells and natural killer cells [90]. These findings highlight the need to determine whether natural infection and vaccination share the same pathophysiology in these cardiopulmonary effects. They further show the extent of unknowns in this area of study and the need for more research.

This study is not without its limitations, the most notable of which is that we employed a narrative review methodology, which is inherently prone to bias. This limitation could have been rectified by performing a systematic review that encompassed multiple databases and had a predefined protocol, including strict inclusion and exclusion criteria. However, given the breadth of literature and vast nature of this topic, we felt that a narrative review would better suit our desired intent. Additionally, the use of case reports and series to support various cardiopulmonary effects of COVID-19 vaccination is another glaring limitation, given their small sample sizes, lack of proof of causation, and inherent bias. Furthermore, our comparative risk analysis relies on published studies outlining incidence and outcomes of cardiopulmonary effects for both vaccine and natural COVID-19 infection, so it is also prone to bias and must be interpreted with caution. We also failed to perform any statistical analysis to support our claims, further representing the need to interpret it with caution.

## 6. Conclusions

To our knowledge, this is the most comprehensive review of the cardiopulmonary effects of COVID-19 vaccination to date. Cardiovascular complications examined include myocarditis and pericarditis, Takotsubo cardiomyopathy, postural orthostatic tachycardia syndrome (POTS), arrhythmias, sudden cardiac death, and cardiac tamponade. Pulmonary complications examined are pulmonary embolism (PE), interstitial lung disease (ILD), idiopathic pulmonary fibrosis (IPF), pneumonia, eosinophilic granulomatosis with polyangiitis, pneumonitis, and pulmonary hypertension. Despite these complications, the risk–benefit analysis still strongly favors vaccination, as these events occur more frequently with natural infection and confer a significantly worse prognosis. Future studies should identify individuals who are at-risk of developing these cardiopulmonary effects of COVID-19 vaccination, allowing for tailored vaccine schedule recommendations.

## Figures and Tables

**Table 1 vaccines-13-00548-t001:** Cardiovascular effects of COVID-19 vaccination and their management.

Effect	Management
Myocarditis and Pericarditis	Supportive with anti-inflammatories, such as NSAIDs, aspirin, colchicine, and steroids Guideline-directed medical therapy with beta-blockers, angiotensin-converting enzyme (ACE) inhibitors or angiotensin II receptor blockers (ARBs), and diuretics for cases with associated heart failure
Takotsubo Cardiomyopathy	Supportive or heart failure medications, such as ACE inhibitors or ARBs and beta-blockers
Postural Orthostatic Tachycardia Syndrome (POTS)	First-line: lifestyle modifications, such as increased fluid and salt intake, exercise, and compression stockings Second-line: pharmacological therapy with ivabradine, corticosteroids, and beta-blockers, amongst other medications
Arrhythmias	Management is arrhythmia-specific
Sudden Cardiac Death	Resuscitation through Advanced Cardiovascular Life Support (ACLS)
Cardiac Tamponade	Pericardiocentesis

**Table 2 vaccines-13-00548-t002:** Pulmonary effects of COVID-19 vaccination and their management.

Effect	Management
Pulmonary Embolism	Thrombolytic therapy, low-molecular-weight heparin (LWMH), and Factor Xa inhibitors, such as apixaban and rivaroxaban
Interstitial Lung Disease	Corticosteroids
Idiopathic Pulmonary Fibrosis	Corticosteroids
Pneumonia	Supportive care and corticosteroids
Eosinophilic Granulomatosis with Polyangiitis	Corticosteroids Cyclophosphamide for more severe cases Plasmapheresis, rituximab, and tumor necrosis factor (TNF) inhibitors can be used for refractory disease
Pneumonitis	Corticosteroids
Pulmonary Hypertension	Anticoagulation for thrombus-associated cases Nitric oxide

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
