# Peer review of "Cardiopulmonary Effects of COVID-19 Vaccination: A Comprehensive Narrative Review"

_vaccines, 2025, doi:10.3390/vaccines13060548_

Round 1

Reviewer 1 Report

Comments and Suggestions for Authors

The article delivers a detailed overview of the cardiac and lung problems related to COVID-19 mRNA vaccines including myocarditis and pulmonary hypertension and pneumonia. It establishes that although these adverse effects may occur, they are relatively rare and that the risk/benefit ratio of vaccination is on the side of vaccination since complications  of natural COVID-19 are more frequent and more severe. The review presents a number of case reports and studies which explain the symptoms, diagnosis and possible reasons for these complications, and stresses the importance of  monitoring patients with pre-existing conditions. The results of the study confirm that vaccination is a vital tool in reducing the risks of severe disease from COVID-19, while also documenting adverse effects.

Minor observations:

It is recommended to the authors to delve deeper on these points in the introduction or other sections:

  • An overview of the COVID-19 pandemic and the global response, including the rapid development and deployment of mRNA vaccines as a key event in public health.
  • Recent studies and findings related to the cardiopulmonary effects of COVID-19 vaccination, noting that this is an area of ongoing research with important implications for patient care.
  • Comparative Analysis: The risks of vaccine associated complications compared with those associated with COVID-19 infection itself should be discussed in order to understand these risks within the context of vaccination strategies.
  • Public Health Implications: This research has important implications for informing public health policy and vaccination recommendations, particularly for vulnerable populations who may be at higher risk for adverse events.

Major observations:

I suggest the authors to provide a brief account of the methods used for information retrieval. Although it is not a systematic literature review, reference management, methodology, databases selected, search terms, and  other factors can greatly affect the quality of a thematic review such as the one presented.

Suggestion:

A figure or graphic would definitely improve the paper, but it is only a suggestion.

Author Response

Dear Reviewer, 

Thank you for the opportunity to revise our manuscript titled: Cardiopulmonary Effects of COVID-19 Vaccination: A Comprehensive Narrative Review.

We appreciate the careful review and constructive suggestions. It is our belief that the manuscript is substantially improved after making the suggested edits. 

Following this letter are your comments with our responses. We look forward to hearing from you in due time regarding our submission and to respond to any further questions and comments you may have. 

From all authors to you, thank you for taking the time to critically review our manuscript! 

Reviewer’s Comments

Authors’ Responses

“It is recommended to the authors to delve deeper on these points in the introduction or other sections:

An overview of the COVID-19 pandemic and the global response, including the rapid development and deployment of mRNA vaccines as a key event in public health.”

We have expanded upon this in the Introduction Section of our manuscript. Please let us know if you would like us to elaborate more or condense it and we will be happy to do so.

“It is recommended to the authors to delve deeper on these points in the introduction or other sections:

Recent studies and findings related to the cardiopulmonary effects of COVID-19 vaccination, noting that this is an area of ongoing research with important implications for patient care.”

We added a Discussion Section to our manuscript, which includes findings from recent studies surrounding this topic. We also note here the ongoing nature of research in this area.

“It is recommended to the authors to delve deeper on these points in the introduction or other sections:

Comparative Analysis: The risks of vaccine associated complications compared with those associated with COVID-19 infection itself should be discussed in order to understand these risks within the context of vaccination strategies.”

We have expanded upon our comparative risk analysis sections considerably for both the cardiovascular and pulmonary effects of COVID-19 vaccination. Please let us know if you would like us to delve deeper and we will be happy to do so.

“It is recommended to the authors to delve deeper on these points in the introduction or other sections:

Public Health Implications: This research has important implications for informing public health policy and vaccination recommendations, particularly for vulnerable populations who may be at higher risk for adverse events.”

We discuss this in the first three paragraphs of our Discussion Section. Additionally, the risks of cardiopulmonary effects by population are addressed in the corresponding section outlining the specific cardiopulmonary effect, as well as in the comparative risk analysis sections. Please let us know if you want us to further elaborate and we will be happy to do so.

“Major observations:

I suggest the authors to provide a brief account of the methods used for information retrieval. Although it is not a systematic literature review, reference management, methodology, databases selected, search terms, and other factors can greatly affect the quality of a thematic review such as the one presented.”

Thank you for this suggestion. We agree that the lack of any description of our methodology was a major limitation. We have added a Methods Section to address this. Please let us know if you would like us to elaborate further or change our methodology in a specific way and we will be happy to do so.

“Suggestion:

A figure or graphic would definitely improve the paper, but it is only a suggestion.”

We have added tables outlining the cardiopulmonary effects of COVID-19 vaccination and their management. Please let us know if you would like us to add to these and we will be happy to do so.

Reviewer 2 Report

Comments and Suggestions for Authors

Cardiopulmonary Effects of COVID-19 Vaccination: A Comprehensive Narrative Review:

The present review is the most comprehensive review on the cardiopulmonary effects of COVID-19 vaccination. The authors have summarized cardiovascular complications that include myocarditis and pericarditis, Takotsubo cardiomyopathy, postural orthostatic tachycardia syndrome, arrhythmias, sudden cardiac death, cardiac tamponade and pulmonary complications.

  1. The review would benefit if the authors include general safety profile of the vaccine in the introduction.

  1. Does the vaccine have side effects for all population or is it a specific population like immunocompromised people, diseased people, people with underlying conditions that experience cardiopulamonary effects of vaccine. The review will benefit if the authors shed light on underlying conditions of people that experience those vaccine side effects.

    3.  The discussion/conclusion would benefit if the author describes the need or importance of understanding those off-target effects of vaccine eg personilzed vaccination strategies could be prioritized for people with cardiac or pulmonary conditions.

  1. Are there any biomarkers for people who are more susceptible to cardiopulmonary effects from covid vaccine?

Discussing the above points in the review will improve the overall review. In general, the review is comprehensive and very important as it gives perspective on vaccine related risks. Researchers and vaccine developers can understand the gaps in mechanistic understanding, and it can guide next-gen vaccine safety evaluation.

Author Response

Dear Reviewer, 

Thank you for the opportunity to revise our manuscript titled: Cardiopulmonary Effects of COVID-19 Vaccination: A Comprehensive Narrative Review.

We appreciate the careful review and constructive suggestions. It is our belief that the manuscript is substantially improved after making the suggested edits. 

Following this letter are your comments with our responses. We look forward to hearing from you in due time regarding our submission and to respond to any further questions and comments you may have. 

From all authors to you, thank you for taking the time to critically review our manuscript! 

Reviewer’s Comments

Authors’ Responses

“1. The review would benefit if the authors include general safety profile of the vaccine in the introduction.”

We have included a general safety profile in the Introduction Section for both the Pfizer and Moderna vaccines, based on original trial results. This can be found in Paragraph 3 of the Introduction Section on Lines 55-67.

“2. Does the vaccine have side effects for all population or is it a specific population like immunocompromised people, diseased people, people with underlying conditions that experience cardiopulamonary effects of vaccine. The review will benefit if the authors shed light on underlying conditions of people that experience those vaccine side effects.”

Thank you for this suggestion. We agree that this is an important consideration for this topic and our article. Unfortunately, data seems to be lacking on specific, at-risk populations for these cardiopulmonary effects of COVID-19 vaccination. We lay out the typical populations for these phenomena when we present them. We also reiterate these findings in our Discussion Section Paragraph 2, Lines 571-580. The remainder of our Discussion also focuses on the importance of identifying individuals at risk of these complications. Please let us know if you would like us to elaborate on any aspects of this and we will be happy to do so.

“3. The discussion/conclusion would benefit if the author describes the need or importance of understanding those off-target effects of vaccine eg personalized vaccination strategies could be prioritized for people with cardiac or pulmonary conditions.”

Thank you for this suggestion. We agree that adding this would significantly strengthen our paper. We have added a Discussion Section, which tried to focus on this understanding and the need to identify at-risk populations. Please let us know if you would like us to elaborate and we will be happy to do so.

“4. Are there any biomarkers for people who are more susceptible to cardiopulmonary effects from covid vaccine?”

This was a great thought, and we thank you for the suggestion. We underwent a deep dive into the literature in search of biomarkers predicting cardiopulmonary effects from COVID-19 vaccination but were unable to find articles addressing this. We have addressed this in our Discussion Section Paragraph 4, Lines 595-615. Please let us know if you would like us to elaborate or if you have any suggested articles to include and we will be happy to do so.

Reviewer 3 Report

Comments and Suggestions for Authors

The manuscript entitled "Cardiopulmonary Effects of COVID-19 Vaccination: A Comprehensive Narrative Review”, elaborated the cardiopulmonary effects of COVID-19 vaccination to date. Despite the good structure of the manuscript, there are some concerns which should be addressed.

  1. It would be better if you can design 2 figures illustrating the most common cardiopulmonary events following COVID-19 vaccination.
  2. Why didn’t you address the clinical management of the complications mentioned?, I think it would be better to add tables discussing this.
  3. Could you elaborate on how you conducted the risk-benefit analysis? What specific data do you have to compare the frequency of complications from vaccination versus those from natural COVID-19 infection?
  4. Based on the initial findings and discussions in your introduction, what future research directions do you propose to better understand the long-term effects of COVID-19 vaccination?
  5. Try to check and cite these recently published articles: Karimi, R., Norozirad, M., Esmaeili, F., Mansourian, M., & Marateb, H. R. (2025). COVID-19 Vaccination and Cardiovascular Events: A Systematic Review and Bayesian Multivariate Meta-Analysis of Preventive Benefits and Risks.International journal of preventive medicine,16, 14. https://doi.org/10.4103/ijpvm.ijpvm_260_24. Maatz, H., Lindberg, E.L., Adami, E.et al.The cellular and molecular cardiac tissue responses in human inflammatory cardiomyopathies after SARS-CoV-2 infection and COVID-19 vaccination. Nat Cardiovasc Res4, 330–345 (2025). https://doi.org/10.1038/s44161-025-00612-6

Author Response

Dear Reviewer, 

Thank you for the opportunity to revise our manuscript titled: Cardiopulmonary Effects of COVID-19 Vaccination: A Comprehensive Narrative Review.

We appreciate the careful review and constructive suggestions. It is our belief that the manuscript is substantially improved after making the suggested edits. 

Following this letter are your comments with our responses. We look forward to hearing from you in due time regarding our submission and to respond to any further questions and comments you may have. 

From all authors to you, thank you for taking the time to critically review our manuscript! 

Reviewer’s Comments

Authors’ Responses

“1. It would be better if you can design 2 figures illustrating the most common cardiopulmonary events following COVID-19 vaccination.”

Thank you for this suggestion. We have added a table for both the cardiovascular and pulmonary complications of COVID-19 vaccination, including their management. Please let us know if you would like us to make further changes and we will be happy to do so.

“2. Why didn’t you address the clinical management of the complications mentioned?, I think it would be better to add tables discussing this.”

We addressed the clinical management for these complications in each corresponding section. We also added this to the tables, as mentioned above. Please let us know if you would like us to elaborate on this and we will be happy to do so.

“3. Could you elaborate on how you conducted the risk-benefit analysis? What specific data do you have to compare the frequency of complications from vaccination versus those from natural COVID-19 infection?”

We have significantly expanded our risk-benefit analysis sections for both the cardiovascular and pulmonary effects of COVID-19 vaccination. In these sections, we sought to compare the incidence and clinical course with outcomes between cases of natural infection and vaccination. We utilized comprehensive studies with incidence and outcome data to make these comparisons. Please let us know if you would like us to make changes to this methodology and we will be happy to do so.

“4. Based on the initial findings and discussions in your introduction, what future research directions do you propose to better understand the long-term effects of COVID-19 vaccination?”

Thank you for this suggestion. We have added a Discussion Section, which outlines needed areas of future research. Please let us know if you would like us to elaborate further and we will be happy to do so.

“5. Try to check and cite these recently published articles: Karimi, R., Norozirad, M., Esmaeili, F., Mansourian, M., & Marateb, H. R. (2025). COVID-19 Vaccination and Cardiovascular Events: A Systematic Review and Bayesian Multivariate Meta-Analysis of Preventive Benefits and Risks.International journal of preventive medicine,16, 14. https://doi.org/10.4103/ijpvm.ijpvm_260_24. Maatz, H., Lindberg, E.L., Adami, E.et al.The cellular and molecular cardiac tissue responses in human inflammatory cardiomyopathies after SARS-CoV-2 infection and COVID-19 vaccination. Nat Cardiovasc Res4, 330–345 (2025). https://doi.org/10.1038/s44161-025-00612-6”

Thank you for suggesting these articles. We have added both to our study. Karimi was added to our Discussion Section in Paragraph 3, Lines 582-588. Maatz was added to our Discussion Section in Paragraph 4, Lines 608-612. Please let us know if you would like us to address these studies elsewhere and we will be happy to do so.

Reviewer 4 Report

Comments and Suggestions for Authors

The manuscript lacks originality. It repackages publicly available data without offering new insight or synthesis. Most of the cited studies are individual case reports, pre-existing systematic reviews, or speculative mechanisms. There is no attempt to analyse, reframe, or critically assess this evidence in a novel way.

Major comments:

  • The paper presents no central question. It reads as a list of events without exploring causal relationships, confounders, or comparative outcomes with sufficient depth.
  • Each subsection repeats similar content: a list of symptoms, time of onset, demographic trends, and a brief mention of pathophysiology. These sections show minimal effort to integrate across conditions or analyse larger patterns.
  • The authors claim this is “the most comprehensive review” of cardiopulmonary effects post-COVID-19 vaccination. This is not supported by evidence. Multiple comprehensive reviews with broader datasets and deeper critical analyses exist (e.g., Heymans & Cooper, 2022; Kuehn, 2022; Voleti et al., 2022). This manuscript adds no new data, frameworks, or unique interpretations.
  • There is heavy reliance on case reports without sufficient mention of their limitations. The review fails to differentiate between correlation and causation. No statistical approach is taken to assess significance, bias, or confounders.
  • The manuscript does not explain how the literature was selected or assessed. There is no methodology section describing search terms, inclusion/exclusion criteria, or quality assessment. This undermines the integrity of the review.

Specific comments:

  • The claim of a “centralized reporting system” being unavailable globally (line 54) is misleading. Systems like VAERS, EudraVigilance, and VigiBase exist and are extensively used.
  • Several repeated phrases and awkward constructs, e.g., “vaccine-associated PE typically presents with dyspnea... but cough and syncope have also been reported” (line 222). “But” implies contrast where none exists.
Comments on the Quality of English Language

The English could be improved to more clearly express the research.

Author Response

Dear Reviewer, 

Thank you for the opportunity to revise our manuscript titled: Cardiopulmonary Effects of COVID-19 Vaccination: A Comprehensive Narrative Review.

We appreciate the careful review and constructive suggestions. It is our belief that the manuscript is substantially improved after making the suggested edits. 

Following this letter are your comments with our responses. We look forward to hearing from you in due time regarding our submission and to respond to any further questions and comments you may have. 

From all authors to you, thank you for taking the time to critically review our manuscript! 

Reviewer’s Comments

Authors’ Responses

“The authors claim this is “the most comprehensive review” of cardiopulmonary effects post-COVID-19 vaccination. This is not supported by evidence. Multiple comprehensive reviews with broader datasets and deeper critical analyses exist (e.g., Heymans & Cooper, 2022; Kuehn, 2022; Voleti et al., 2022). This manuscript adds no new data, frameworks, or unique interpretations.”

We have cited these studies and their findings in our article. We also expanded upon our comparative risk analysis sections and added a Discussion Section, which we hope provides some unique interpretations. Please let us know if you have further recommendations to build upon our study and we would be happy to incorporate those.

“There is heavy reliance on case reports without sufficient mention of their limitations. The review fails to differentiate between correlation and causation. No statistical approach is taken to assess significance, bias, or confounders.”

Thank you for pointing out this limitation. We have added a limitations paragraph in our Discussion Section, which can be found in Paragraph 5, Lines 616-628. Please let us know if you think we should expand on this and we would be happy to do so.

“The manuscript does not explain how the literature was selected or assessed. There is no methodology section describing search terms, inclusion/exclusion criteria, or quality assessment. This undermines the integrity of the review.”

Thank you for pointing this out to us. We have added a Methods Section outlining our methodology. Please let us know if you feel further changes are necessary and we would be happy to incorporate these.

“The claim of a “centralized reporting system” being unavailable globally (line 54) is misleading. Systems like VAERS, EudraVigilance, and VigiBase exist and are extensively used.”

Thank you for pointing this out. We agree with you and have removed this.

“Several repeated phrases and awkward constructs, e.g., “vaccine-associated PE typically presents with dyspnea... but cough and syncope have also been reported” (line 222). “But” implies contrast where none exists.”

Thank you for bringing this to our attention. We have updated the wording here.